# Clebsch-Gordan Transformers: Fast and Global Equivariant Attention

## Abstract

The global attention mechanism is one of the keys to the success of transformer architecture, but it incurs quadratic computational costs in relation to the number of tokens. On the other hand, equivariant models, which leverage the underlying geometric structures of problem instance, often achieve superior accuracy in physical, biochemical, computer vision, and robotic tasks, at the cost of additional compute requirements. As a result, existing equivariant transformers only support low-order equivariant features and local context windows, limiting their expressiveness and performance. This work proposes Clebsch-Gordan Transformer, achieving efficient global attention by a novel Clebsch-Gordon Convolution on SO(3) irreducible representations. Our method enables equivariant modeling of features at all orders while achieving $O(N \log N)$ input token complexity. Additionally, the proposed method scales well with high-order irreducible features, by exploiting the sparsity of the Clebsch-Gordon matrix. Lastly, we also incorporate optional token permutation equivariance through either weight sharing or data augmentation. We benchmark our method on a diverse set of benchmarks including n-body simulation, QM9, ModelNet point cloud classification and a robotic grasping dataset, showing clear gains over existing equivariant transformers in GPU memory size, speed, and accuracy.

## 1 Introduction

Transformer-based models have demonstrated effectiveness beyond language processing, showing strong performance in geometry-aware tasks such as robotics, structural biochemistry, and materials science (Wu et al., 2024; Goyal et al., 2023; Pan et al., 2021; Zeni et al., 2024; Rhodes et al., 2025). For instance, 3D robotic perception tasks ranging from segmentation to object matching process point clouds and LiDAR data using attention mechanisms. These tasks heavily rely on token-based representations, and their performance is often constrained by the number of tokens the model can effectively handle. AlphaFold (Jumper et al., 2021), for example, employs equivariant transformers to predict protein structures with unprecedented accuracy by explicitly leveraging SE(3) symmetries such as rotations and translations. However, implementing an equivariant neural network structure typically incurs significant computational overhead and increased inference time. As a result, most current approaches are limited to small symmetry groups or low-order representations (Thomas et al., 2018; Fuchs et al., 2020; Moskalev et al., 2024; Liao and Smidt; Satorras et al., 2022). Enabling fast, low-memory overhead equivariant operations over large context windows is essential to scaling robust and sample-efficient learning in geometry-aware domains.

Unfortunately, maintaining equivariance while modeling a global geometric context is challenging due to the computational demands of processing high dimensional data at scale. There are essentially two components that contribute to the computational complexity of $E(3)$-equivariant transformers: the time and memory scaling of the transformer with the number of tokens, $N$, and the time and memory complexity on the maximum harmonic degree, $\ell$. Naively, a global equivariant attention mechanism will have $O(N^2)$ token complexity and $O(\ell^6)$ harmonic complexity (Passaro and Zitnick, 2023b). By assuming only local attention, the token complexity can be reduced to $O(dN)$, where $d$ is the local context window, at the cost of discarding information about long range correlations. In addition,

various approximation techniques have been used to reduce the harmonic complexity to $O(\ell^3)$ (Luo et al., 2024). Recently, SE(3)-Hyena achieved $O(N \log N)$ computational complexity using long convolution (Romero et al., 2021; Poli et al., 2023) in the Fourier domain. However, it only supports up to first-order irreducible representations of SO(3) (i.e., scalar and vectors), making it difficult to capture higher-degree angular dependencies and limiting its ability to represent more complex, structured geometric patterns. Moreover, SE(3)-Hyena is not permutation invariant, making it most suitable for point clouds with a natural ordering.

This raises the question: can we design a method with global equivariant attention and $\mathcal{O}(N \log N)$ token complexity, support for arbitrary orders of spherical harmonics with $\mathcal{O}(\ell^3)$ complexity, and permutation invariance? This work addresses this challenge by introducing Clebsch-Gordon Convolution on SO(3) irreducible representations of arbitrary degree. We also enforce or encourage permutation invariance through either weight sharing or data augmentation. Our contributions can be summarized as follows:

- We generalize the SE(3)-Hyena method of (Moskalev et al., 2024) to include equivariant features of all types. By exploiting the sparsity of the Clebsch-Gordon matrix, our method achieves $O(L^3)$ harmonic scaling.

- By applying our proposed attention in the graph spectral domain, we achieve permutation-equivariant global attention in $O(N \log N)$ time.

- We benchmark our method on a diverse array of tasks, including robotics, computer vision, and molecular biochemistry. Our method outperforms current state of the art methods on all tasks, with considerable reduction in memory usage.

## 2 RELATED WORK

**SE(3)-Equivariance:** SE(3)-equivariant neural networks have three main classes: (1) those based on group convolution (Cohen and Welling, 2016), which discretizes SE(3), transforms a convolutional filter according to each group element in the discretization, and lastly performs cross-correlation using the transformed convolutional filter (Cesa et al., 2022b; Chen et al., 2021; Zhu et al., 2023); (2) those based on irreducible (spherical Fourier) representations, which provide a compact representation of SO(3) signals at each point in the point cloud (Thomas et al., 2018; Brandstetter et al., 2022; Liao and Smidt; Fuchs et al., 2020; Passaro and Zitnick, 2023a; Liao et al., 2024), and (3) those based on scalar, vector, and multivector representations (Deng et al., 2021; Moskalev et al., 2024; Brehmer et al., 2023). Compared with group convolution (class (1)), the irreducible spherical Fourier representation (class (2)) is more compact and avoids discretization errors—among these works, Tensor Field Network (TFN) Thomas et al. (2018) leverages the tensor product to propagate information between points, SE(3)-Transformer Fuchs et al. (2020) extends TFN by using attention in the Fourier domain for local information passing, and ESCN Passaro and Zitnick (2023a) proposes approximating the tensor product in SO(3) by that in $SO(2)$, significantly reducing computational complexity. Vector representation (class (3)) has limited expressiveness, while the irreducible representation improves expressiveness as its order increases Liao and Smidt. This work achieves efficient SE(3)-equivariance with high-order irreducible representations by introducing a linear-time attention mechanism based upon the vector long convolution introduced by Moskalev et al. (2024).

**Subquadratic Attention:** Several linear-time attention mechanisms have been proposed to overcome the quadratic time and memory complexity of standard Transformer architectures. Reformer Kitaev et al. (2020) utilizes locality-sensitive hashing (LSH) to approximate self-attention. Choromanski et al. (2021) replaces the standard softmax attention with a kernel-based approximation called FAVOR+ (Fast Attention Via positive Orthogonal Random features), achieving linear time and space complexity. Nyströmformer Xiong et al. (2021), leverages the Nyström method to approximate the self-attention matrix using a set of landmark points. Linformer Guo et al. (2024) addresses the quadratic memory and computation bottleneck of standard Transformers by approximating self-attention with low-rank projections. These methods enable LLM transformers to process much longer sequences than previously feasible.

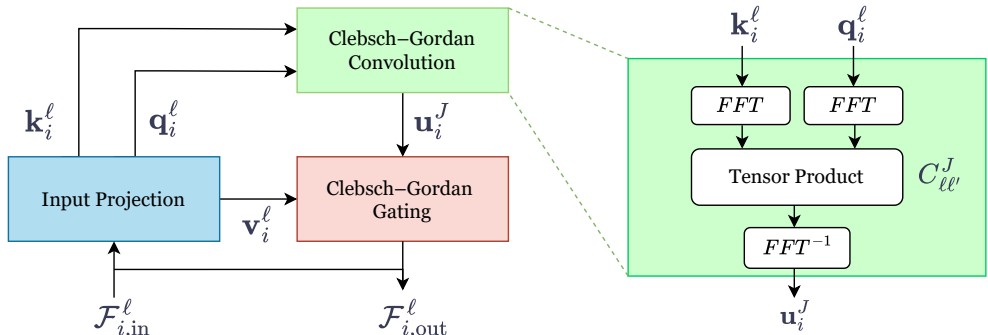

Figure 1: Schematic of Clebsch-Gordon Convolution. Left: Inputs $f_i^\ell$ are projected into queries $q_i^\ell$, keys $k_i^\ell$, and values $v_i^\ell$ using an $SE(3)$-equivariant projection layer. Queries and keys are passed into a Clebsch-Gordon Convolution which outputs $u_i^\ell$ with which a tensor product of values is computed. Outputs are added with a residual connection. Right: Queries $q_i^\ell$ and keys $k_i^\ell$ are processed using Clebsch-Gordon Convolution. Queries and keys are first fast Fourier transformed, then a tensor product is applied. The output is fast Fourier transformed back.

## 3 Background

**Attention.** Attention is a data-dependent linear map (Vaswani et al., 2023) describing the pairwise interaction between tokens in a transformer's input context. Let $q_i$, $k_i$ and $v_i$ be linear projections of input. Attention is defined $\text{Attn}(q, k, v) = \text{softmax}[\alpha(q, k)]v$ where $\alpha(q, k)_{ij} = q_i^T k_j$ is the attention matrix. The computation of $\alpha$ scales quadratically in the input size, which is the main bottleneck in the transformer architecture in large models. Numerous methods attempt to compute $\alpha(q, k)$ faster using some numerical approximation.

**Equivariant Attention and Message Passing.** Numerous equivariant methods attempt to generalize attention to process equivarient features. The two most common forms are equivariant attention, or equivariant message passing (Brandstetter et al., 2022). In the standard setup, the $i$-th graph node is located at position $x_i \in \mathbb{R}^3$ has features $f_i^\ell$ where the index $\ell$ specifies feature type. Attention and message passing then attempt to process information via update rules

$$\text{Attention: } f_{out,i}^\ell = W_V^{\ell\ell} f_{in,i}^\ell + \sum_{k=0}^{L} \sum_{j \in \mathcal{N}_i} \alpha_{ij} W_V^{\ell k}(x_i - x_j) f_{in,j}^k \tag{1}$$

$$\text{Message Passing: } f_{out,i}^\ell = \phi(f_{in,i}^\ell, \sum_{j \in \mathcal{N}_i} m_{ij}), \quad m_{ij} = \psi(f_{in,i}^\ell, f_{in,j}^{\ell'}, \|x_i - x_j\|) \tag{2}$$

where $\mathcal{N}_i$ is some neighborhood of points around point $j$.

Memory constraints force methods like (Fuchs et al., 2020; Passaro and Zitnick, 2023b; Satorras et al., 2022; Brandstetter et al., 2022; Thomas et al., 2018) to restrict to small neighborhoods $\mathcal{N}_i$ of size less than 50. We show in Sec. 5 that decreasing the size of the local context window can lead to significant changes in performance. A recent method (Moskalev et al., 2024) showed that for invariant and vector convolutions adding global context can improve model performance. We extend this work to equivariant features of all types.

## 4 Method

Let $F^n = \{f_i^\ell\}_{i=1}^n$ be a set of $n$ input features to an SO(3)-equivariant transformer, where SO(3) acts upon each feature $f_i^\ell \in \mathbb{R}^{(2\ell+1) \times m_\ell}$ via its $\ell^{\text{th}}$ irreducible representation. We denote the multiplicity (i.e., channel dimension) of the input of type $\ell$ as $m_\ell$. From the

Table 1: Key properties of equivariant attention mechanisms. $N$: number of tokens, $d$: average graph degree, $L$: maximum harmonic degree.

| Model | Global Attn. | Perm. Equiv. | Token Complex. | Harmonic Complex. |
|---|---|---|---|---|
| SEGNN (Brandstetter et al., 2022) | ✗ | ✓ | $\mathcal{O}(dN)$ | $\mathcal{O}(L^6)$ |
| SE(3)-Transformer (Fuchs et al., 2020) | ✗ | ✓ | $\mathcal{O}(dN)$ | $\mathcal{O}(L^6)$ |
| Equiformer-v2 (Passaro and Zitnick, 2023b) | ✗ | ✓ | $\mathcal{O}(dN)$ | $\mathcal{O}(L^3)$ |
| SE(3)-Hyena (Moskalev et al., 2024) | ✓ | ✗ | $\mathcal{O}(N \log N)$ | Type 0 & Type 1 only |
| **Ours** | ✓ | ✓/learned | $\mathcal{O}(N \log N)$ | $\mathcal{O}(L^3)$ |
| Theoretical Ideal | ✓ | ✓ | $\mathcal{O}(N \log N)$ | $\mathcal{O}(L^2 \log L)$ |

features in $F^n$, we encode queries, $Q_{F^n}$, keys, $K_{F^n}$, and values, $V_{F^n}$, as:

$$q_i^\ell = W_Q^\ell(f_i^{\ell'}), \quad k_i^\ell = \sum_{\ell'} W_K^\ell(f_i^{\ell'}), \quad v_i^\ell = W_V^\ell(f_i^{\ell'})$$

where $W_Q^\ell$, $W_K^\ell$, and $W_V^\ell$ are learnable equivariant mappings converting type $\ell'$ features of multiplicity $m_\ell'$ into type $\ell$ features of multiplicity $m_\ell$.

Our proposed method is agnostic to the particular equivariant encoders used; see appendix E.2 for more details. We seek to compute self-attention over $Q_{F^n}$, $K_{F^n}$, and $V_{F^n}$ in linear time, scaling to global context, and—unlike earlier work (Moskalev et al., 2024)— remaining compatible with $F^n$ comprising equivariant features of any type. Our proposed method extends the core idea of Poli et al. (2023), building upon the vector long convolution introduced by Moskalev et al. (2024).

### 4.1 CLEBSCH-GORDON CONVOLUTION

We structure our attention mechanism as follows: first, inspired by Moskalev et al. (2024), we define the following operation, where $C_{\ell\ell'}^J$ is the Clebsch-Gordan matrix projecting from features of type $\ell \otimes \ell'$ onto features of type $J$:

$$(q^\ell \star k^{\ell'})_i^J = C_{\ell\ell'}^J \sum_{j=1}^N q_j^\ell \otimes k_{i-j}^{\ell'} \tag{3}$$

which takes as input features of types $\ell$ and $\ell'$ and outputs features of type $J$. If $q_i^\ell \in \mathbb{R}^{(2\ell+1)m_\ell}$ and $k_i^\ell \in \mathbb{R}^{(2\ell+1)m_\ell}$ the resultant tensor product has dimension $(q^\ell \star k^{\ell'})_i^J \in \mathbb{R}^{(2J+1)m_\ell m_{\ell'}}$. In practice, we found that using multiple heads (which do not interact during the tensor product) led to better performance; see appendix E for further discussion. Operations of the form $C_{\ell\ell'}^J q^\ell \otimes k^{\ell'}$ are ubiquitous in machine learning, making their fast computation a subject of great research interest. For the special case when the keys are spherical harmonic outputs, i.e., $k^\ell = Y^\ell(\hat{n})$, Passaro and Zitnick (2023b) used a group theoretic decomposition to reduce SO(3) operations into SO(2) operations. Luo et al. (2024) generalized this idea and used the Gaunt tensor product coefficients to reduce the tensor product computation to a highly tractable two dimensional Fourier transformation for general input features. We compute the tensor product in eq. (3) using a slight modification of the methods proposed in (Luo et al., 2024); see appendix A for details. The operation in eq. (3), picks the type $J$ output out of the tensor product. By definition of the Clebsch-Gordan matrix $C_{\ell\ell'}^J$, the tensor product of type $\ell$ and type $\ell'$ features decomposes as

$$(q_i^\ell \otimes k_{i-j}^{\ell'}) = \bigoplus_J C_{\ell\ell'}^J (q_i^\ell \otimes k_{i-j}^{\ell'})^J \tag{4}$$

where $\bigoplus$ is the direct sum of vector spaces. To allow all query and key types to interact, we want to compute, for each $J$, $\hat{u}_i^J = \sum_{\ell\ell'} (q_i^\ell \star k_i^{\ell'})^J$. Following the notation of Luo et al. (2024), let $\tilde{q}_i = [q_i^0, q_i^1, ..., q_i^L]$ be the stack of all query vectors containing irreducibles of up

to degree $L$ and let $\tilde{k}_i = [k_i^0, k_i^1, ..., k_i^L]$ be the stack of all keys containing irreducibles of up to degree $L$. The full tensor product of these features for output type $J$ is given by

$$u_i^J = (\tilde{q}_i \star \tilde{k}_i)^J = \sum_{\ell=1}^{L} \sum_{\ell'=1}^{L} (q_i^\ell \star k_i^{\ell'})^J$$

Naively retaining all output-irreducible types of the Clebsch-Gordan tensor product up to type $L$ requires $O(L^3)$ 3D matrix multiplications, for a total complexity of $O(L^6)$. We then define the full convolution as $\tilde{u}_i^J = (\tilde{q}_i \star \tilde{k}_i)$. This convolution computation can be simplified in two ways. The Fourier transform $\hat{u}_q^\ell$ of $u_i^\ell$ over the spatial index can be written as $\hat{u}_i^J = C_{\ell\ell'}^J \hat{q}_i^\ell \otimes \hat{k}_i^{\ell'}$ which is a matrix multiplication in Fourier space. Using the Fast Fourier Transform, the computation of $\hat{q}^\ell$ and $\hat{k}^\ell$ can be done in time $O(N \log N)$. We further consider both intra-channel and inter-channel tensor products; see appendix E for additional ablation studies.

### 4.1.1 INVARIANT GATING

A key aspect of the transform proposed in (Moskalev et al., 2024) is its non-linear data dependent gating. Accordingly, after obtaining $\hat{u}_q^J$, we compute a set of invariant features $I^\ell = \gamma^\ell(\hat{u}^0, \hat{u}^1, \hat{u}^2, ...)$ with one $I^\ell$ for each irreducible type $\ell$. We evaluated a variety of encoder types for $\gamma^\ell$; see appendix E.4 for ablation studies. The gating $I^J$ is of dimension $N \times m_J$. We then apply softmax gating $u_i^J \to \sigma(I^\ell)u_i^J$ and combine the resultant gated features $u_i^J$ with the values via another tensor product and Clebsch-Gordon matrix projection

$$f_{i,out}^J = \sum_{\ell\ell'} C_{\ell\ell'}^J u_i^\ell \otimes v_i^{\ell'} \tag{5}$$

Note that the second multiplication, eq. (5) is done in real space rather than Fourier space. The idea of switching between real and Fourier space when computing attention is a key idea developed by Poli et al. (2023); Stachenfeld et al. (2020), allowing the model to fuse both global and local information. Lastly, we apply an equivariant MLP to reduce the multiplicity dimension back down to that of the input $\hat{f}_i^\ell \to \text{MLP}(\hat{f}_i^\ell)$. We then add the attention features to the input features as a residual via

$$f_{i,out}^\ell = f_{i,in}^\ell + \text{MLP}(f_{i,in}^\ell)$$

By subtracting off and re-adding the mean of inputs $f_i^\ell$, the outputs $\hat{f}_{i,out}^\ell$ are be fully SE(3)-equivariant.

---

**Algorithm 1:** Clebsch-Gordon Convolution

**Input** : Input signal $f_{i,in}^\ell$
**Output** : Output Attention $f_{i,out}^J$
1 **Encode:**
   $q_i^\ell = W_Q^\ell(f_i^{\ell'}), k_i^\ell = W_K^\ell(f_i^{\ell'}), v_i^\ell = W_V^\ell(f_i^{\ell'})$
2 **FFT:** $q_i^\ell, k_i^\ell \to \hat{q}_i^\ell, \hat{k}_i^\ell$
3 **Tensor Product:** $u_i^J = C_{\ell\ell'}^J \hat{q}_i^\ell \otimes \hat{k}_i^{\ell'}$
4 **Inverse FFT:** $\hat{u}_i^\ell \to u_i^\ell$
5 **Tensor Product:** $f_{i,out}^J = C_{\ell\ell'}^J u_i^\ell \otimes v_i^{\ell'}$
6 **return** $f_{i,out}^J$

---

In practice, we found that using a head dimension of 4 or 8 with a maximum harmonic of $L = 5$ of $L = 6$ and a channel dimension of 8 or 16 was optimal. See appendix E.3 for ablations on model parameters. In general, we found that using a fixed local context window allowed for much greater performance. Specifically, out full output features are

$$f_{i,out}^\ell = \mathcal{F}^{CG}(f_{i,in}^\ell) + \mathcal{F}^{SGNN}(f_{i,in}^\ell)$$

where $\mathcal{F}^{CG}(f_{i,in}^\ell)$ is the output of the Clebsch-Gordon convolution and $\mathcal{F}^{SGNN}(f_{i,in}^\ell)$ is a standard equiformer layer (Passaro and Zitnick, 2023b), with a fixed local context window. We conjecture using both global and local attention that this allows the model to better process both local and global information. This idea is inspired by Han et al. (2024), where it is conjectured that state space models combined with some small local attention can capture both global and local features.

### 4.2 ARCHITECTURE

Figure 1 shows the full flow of our proposed attention block. Specific choices for architecture for experiments is discussed more in E.

### 4.3 Sparsity of the Clebsch-Gordon Matrix

We now turn to improving the performance of our proposed attention mechanism by exploiting sparsity properties of the Clebsch-Gordon matrix. Consider the operation defined by $u^J = \sum_{\ell\ell'} C^J_{\ell\ell'} q^\ell \otimes k^{\ell'}$. The naive cost of this operation is $\mathcal{O}((2J+1)(2\ell+1)(2\ell'+1))$. Thus, the total computation of $u^\ell$ for each $\ell$ is naively $O(J\ell\ell') = JL^4$ where $L$ is the maximum harmonic used. Fortunately, this neglects the sparsity of the matrix $C^J_{\ell\ell'}$. Specifically, $C^J_{\ell\ell'}$ is a tensor of size $(2J+1) \times (2\ell+1) \times (2\ell'+1)$ with most elements equal to zero. To see this, let $|\ell m\rangle$ be the basis for the $\ell$ representation. Then, the Clebsch-Gordon coefficients $C^{JM}_{\ell m \ell' m'}$ satisfy

$$|JM\rangle = \sum_{mm'} C^{JM}_{\ell m \ell' m'} |jm\rangle \otimes |j'm'\rangle$$

because $J^2$ and $J_z$ can be simultaneously diagonalized, it is always possible to find a basis (e.g., the $z$-basis is standard convention in physics) where $J_z|jm\rangle = m|jm\rangle$ applying this relation to the definition of the Clebsch-Gordon coefficients, we have that

$$J_z|JM\rangle = M|JM\rangle \implies M|JM\rangle = \sum_{mm'} C^{JM}_{\ell m \ell' m'}(m + m')|\ell m\rangle \otimes |\ell' m'\rangle$$

Ergo, the matrix elements $C^{JM}_{\ell m \ell' m'}$ are non-zero only when $M = m + m'$. In physicists language, the sparsity of $C^{JM}_{\ell m \ell' m'}$ is a *selection rule* that follows from conservation of the z-component of angular momentum. Thus, the total number of non-zero elements in $C^J_{\ell\ell'}$ is $(2\ell+1)(2\ell'+1)$—much smaller than the naive estimate of $(2J+1) \times (2\ell+1) \times (2\ell'+1)$. Exploiting this fact, we can compute the tensor product in time dependent only on the input sizes. Parity symmetry further requires that any $C^{JM}_{\ell m, \ell' m'}$ where $\ell + \ell' + J$ is odd is identically zero; it also constrains some elements to be redundant. We compare the sparse method with the Gaunt tensor product method of Luo et al. (2024) and the e3nn implementation (Geiger and Smidt, 2022) in appendix E.

### 4.4 Special Case: Vector Long Convolution

We show that our method recovers the vector long convolution method proposed in (Moskalev et al., 2024) as a special case. This vector long convolution is a fast $O(N \log N)$ attention mechanism for queries, keys and values of type 1. Let $q_i$ and $k_i$ be type 1 queries and keys. The vector convolution from (Moskalev et al., 2024) is defined as

$$(q \star k)_i = \sum_j q_j \times k_{i-j} \tag{6}$$

where $\times$ denotes the standard vector cross product. Elementwise, the vector convolution has $(q \star k)_{ia} = \epsilon_{abc} \sum_j q_{jb} k_{(i-j)c}$. The vector convolution $q \star k$ transforms in the vector representation of $SO(3)$. Equation (6) is equal to the expression $(q \star k)_i = C^1_{11} \sum_j q_j \otimes k_{(i-j)}$. To see, this note that the tensor $C^1_{11}$, which is of dimension $(3 \times 3 \times 3)$ can be written out as a $(3 \times 9)$ matrix with elements given by

$$C^1_{11} = \begin{bmatrix} 0 & 0 & 0 & 0 & 0 & \frac{1}{\sqrt{2}} & 0 & \frac{1}{\sqrt{2}} & 0 \\ 0 & 0 & \frac{1}{\sqrt{2}} & 0 & -\frac{1}{\sqrt{2}} & 0 & \frac{1}{\sqrt{2}} & 0 & 0 \\ 0 & \frac{1}{\sqrt{2}} & 0 & \frac{1}{\sqrt{2}} & 0 & 0 & 0 & 0 & 0 \end{bmatrix}$$

Note the for two 3-vectors $q$ and $k$, this can be written as $C^1_{11}(q \otimes k) = \frac{1}{\sqrt{2}}(q \times k)$. Thus, for any three vectors $C^1_{11}(q \otimes k) = \frac{1}{\sqrt{2}}(q \times k)$ is proportional to the standard cross product. Thus, we have that

$$(q \star k)_{ia} = \epsilon_{abc} \sum_j q_{jb} k_{(i-j)c} = \sqrt{2} C^1_{11} \sum_j q_j \otimes k_{(i-j)}$$

which reduces to our method for input types $(1,1)$ and output type 1. Thus, the method proposed in (Moskalev et al., 2024) can be seen as a special case of our method when tensor product input features are restricted to be pairs of invariant or vector features.

Table 2: N-body simulation results ($N = 5$ particles). Mean squared error for position ($x$) and velocity ($v$) prediction. Literature values in parentheses. NR = not reported.

| Model | Linear | Ours | Ours w/o local | SE(3)-Hyena | Set Trans. | SE(3)-Trans. | SEGNN | EGNN | TFN |
|---|---|---|---|---|---|---|---|---|---|
| MSE $x$ | 6.91e-2 | **4.1e-3**±3e-4 | 5.0e-3±3e-4 | 7.1e-3 (1.8e-3) | 1.39e-2 | 7.6e-3 (7.6e-3) | 4.8e-3 (5.6e-3) | 7.0e-3 | 1.50e-2 |
| ΔEQ | — | 9.6e-6 | 1.0e-5 | 1.1e-4 | 1.67e-1 | 3.2e-7 | NR | NR | NR |
| MSE $v$ | 2.61e-1 | **6.5e-3**±2e-4 | 7.5e-3±3e-4 | 7.1e-3±7e-4 | 1.01e-1 | 7.5e-2 (7.5e-2) | NR | NR | NR |
| ΔEQ | — | 4.8e-7 | 5.2e-7 | 1.2e-6 | 3.70e-1 | 6.3e-7 | NR | NR | NR |

## 5 EXPERIMENTS

### 5.1 BASELINES

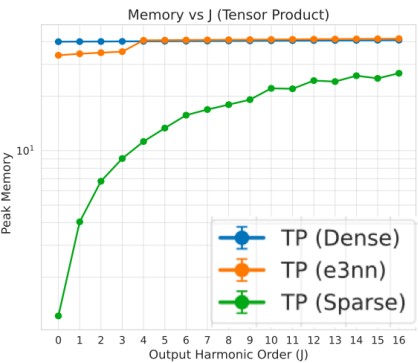

Figure 2: Memory usage versus the number of irreducible representations $J$ for our tensor product attention mechanism. We compare with dense matrix multiplication and e3nn (Geiger and Smidt, 2022).

See appendix A for all experiments.

We compare our method with state of the art baselines for point cloud processing. The SE(3)-transformer (Fuchs et al., 2020) is an equivariant attention mechanism based on Tensor Field Networks (Thomas et al., 2018). Equiformer v2 (Liao et al., 2024) uses a convolutional trick from Passaro and Zitnick (2023b) to reduce the harmonic complexity to $O(L^3)$ from $O(L^6)$. Fused SE(3)-transformer implements the method of Fuchs et al. (2020) using fused kernels for decreased computational overhead. SE(3)-Hyena (Moskalev et al., 2024) uses a modification of the Hyena architecture to do global linear time attention on invariant (type 0) and vector (type 1) features. The use of only invariant and vector features significantly limits model expressivity.

We benchmark our method on the ModelNet40 (Wu et al., 2015) classification task, Nbody trajectory simulation (Fuchs et al., 2020), QM9 (Ramakrishnan et al., 2014), and a custom robotic grasping dataset.

### 5.2 NBODY SIMULATION

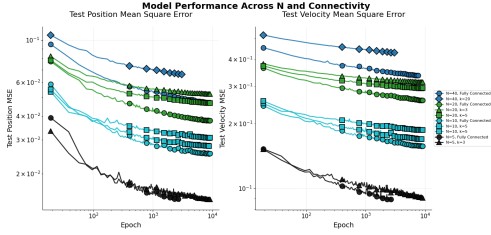

Figure 3: Performance of $SE(3)$-transformer on the Nbody dataset for different number of points $N$ and nearest neighbors $k$. We use the exact model parameters at that of Fuchs et al. (2020). Note that on the $N = 20$ curves decreasing $k = 20$ (fully connected graph) to $k = 5$ leads to over a twenty percent drop in performance. Left shows MSE of position, right shows MSE of velocity. Local context window based methods fail to capture accuracy as tasks get more difficult; global context is needed.

We first test on method on the Nbody simulation task, described in Fuchs et al. (2020). In the simulation in Fuchs et al. (2020), five particles each carry either a positive or a negative charge and exert repulsive or attractive forces on each other. The network input is the position of a particle in a specific time step, its velocity, and its charge. The algorithm must predict the relative location and velocity 500 time steps into the future.

For this test, our model consists of a equivariant graph convolution (Brandstetter et al., 2022), followed by 4 equivariant Clebsch-Gordan attention layers. We use irriducibles types up to six, each with channel dimension of 8 and head dimension of four. Models were trained for 500 epochs, using cosine annealing scheduler with an initial learning rate of $1e^{-3}$ and gradient clipping. Each run was run on a single NVIDIA V100 GPU.

Table 3: Performance comparison for varying numbers of particles $N$ and nearest neighbors $k$ (fc = fully connected graph). Our method uses fixed $k = 3$ local attention. Averaged over 2 seeds.

| Method | N = 5 | | N = 10 | | | N = 20 | | | | N = 40 | | | | |
|---|---|---|---|---|---|---|---|---|---|---|---|---|---|---|
| | $k$=3 | fc | $k$=3 | $k$=5 | fc | $k$=3 | $k$=5 | $k$=10 | fc | $k$=3 | $k$=5 | $k$=10 | $k$=20 | fc |
| SE(3)-Transformer | 0.013 | 0.013 | 0.031 | 0.028 | 0.025 | 0.057 | 0.052 | 0.050 | 0.044 | 0.061 | 0.056 | 0.052 | 0.049 | OOM |
| SEGNN | 0.040 | 0.048 | 0.023 | 0.018 | 0.013 | 0.042 | 0.039 | 0.033 | 0.029 | 0.052 | 0.480 | 0.450 | 0.038 | OOM |
| Ours w/o local | — | **0.003** | — | — | 0.012 | — | — | — | 0.026 | — | — | — | — | 0.031 |
| Ours | — | **0.003** | — | — | **0.010** | — | — | — | **0.023** | — | — | — | — | **0.030** |

Table 4: Mean absolute error (MAE) on QM9 dataset. Lower values indicate better performance.

| Model | Mean Absolute Error | | | | | |
|---|---|---|---|---|---|---|
| | $\alpha$ (Bohr$^3$) | $\Delta\epsilon$ (meV) | $\epsilon_{HOMO}$ (meV) | $\epsilon_{LUMO}$ (meV) | $\mu$ (D) | $C_v$ (cal/mol·K) |
| SE(3)-Transformer | 0.142 | 53.0 | 35.0 | 33.0 | 0.510 | 5.4e-2 |
| EGNN | 7.1e-2 | 48.0 | 29.0 | 25.0 | 0.290 | 3.1e-2 |
| SEGNN | 6.0e-2 | 42.0 | 24.0 | 21.0 | 2.3e-2 | 0.310 |
| Ours w/o local | 0.100 | 49.0 | 31.0 | 26.0 | 0.310 | 0.350 |
| **Ours** | 0.100 | **39.0** | **26.0** | **19.0** | **0.210** | **3.0e-2** |

As motivation, we show that the model performance can vary with the size of the message passing window. Specifically, in the nbody simulations of Fuchs et al. (2020); Brandstetter et al. (2022); Satorras et al. (2022) an all to all connection is used. This will scale quadratically in the number of particles, which quickly becomes computationally untractable. We 3 perform an ablation study on Fuchs et al. (2020) and Brandstetter et al. (2022) where we use a $k$-nearest neighbors, as opposed to fully connected graphs.

As is shown in table 3, our method achieves state of the art performance on this task. Furthermore, our model is highly scalable to larger $N$. Although this is a toy task, we believe these results illustrates both the scalability of our method and the danger of using methods with only local context. Additional experiments and ablations are in appendix A.

## 5.3 QM9

For molecular chemistry, we benchmark on the QM9 dataset (Ramakrishnan et al., 2014), a widely-used collection of 134k small organic molecules with up to 9 heavy atoms (C, O, N, F). Each molecule is annotated with 19 regression targets, including atomization energies, dipole moments, and HOMO-LUMO gaps, calculated using DFT. Following standard practice, we predict one target at a time using the provided 110k/10k/10k training/validation/test split, and report mean absolute error (MAE) in units consistent with prior work.

In its current form, our model does not use edge features. For that reason, we first apply an SEGNN layer (Brandstetter et al., 2022) with skip connection, followed by four of our attention blocks. Output invariant features are then fed into an MLP for classification. For this test, our model consists of an equivariant graph convolution (Brandstetter et al., 2022), followed by 8 equivariant Clebsch-Gordan attention layers. We use irreducibles up to $\ell = 6$, each with 8 channels and 4 heads. Models were trained for 500 epochs, using cosine annealing scheduler with an initial learning rate of $1e^{-4}$ and gradient clipping. Each run was run on eight NVIDA V100 GPUs. Additional experiments and ablations are shown in appendix A.

## 5.4 MODELNET40 CLASSIFICATION

The ModelNet40 classification task is a widely used benchmark for evaluating 3D shape recognition methods. Introduced by Wu et al. (2015), the ModelNet40 dataset consists of $12,311$ 3D CAD models from 40 object categories. The dataset is split into $9,843$ training examples and $2,468$ test examples. The task is to classify 3D objects based on geometric

Table 5: Classification accuracy on ModelNet10 and ModelNet40. All models trained with scale, rotation, and permutation augmentation.

| Model | Year | MN10 Acc.(%) | MN40 Acc.(%) |
|---|---|---|---|
| DGCNN | 2019 | 95.1 | 92.9 |
| PointNet++ | 2020 | 97.4 | 91.3 |
| SEGNN | 2021 | 94.2 | 90.5 |
| SE(3)-Trans. | 2020 | 93.2 | 88.1 |
| Ours w/o local | 2025 | 95.5 | 89.3 |
| Ours | 2025 | 90.1 | 85.9 |

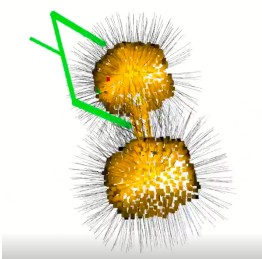

Figure 4: Robotic object dataset example showing an object with surface normals and optimal grasp locations.

structure. We compare our model with DGCNN (Wang et al., 2019) and PointNet++ (Qi et al., 2017) which are state of the art non-equivariant methods.

### 5.5 Object Grasping Dataset

We also consider a bespoke robotic grasping dataset. Robotic grasping fundamentally depends on object geometry, and high precision robotic grasping is difficult because it requires both fine angular resolution and large context window. Our dataset consists of 400 samples, each of which consists of a point cloud, a set of surface normal vectors, an optimal grasp orientation (represented as a $3 \times 3$ matrix), a optimal grasp depth (which is a single positive real number) and an optimal grasp location. Each point cloud has resolutions of 512, 1024, 2048, or 4096 points. We consider three tasks for the object grasping dataset; namely, surface normal prediction and grasp prediction. We detail these tasks in B.

Table 6: Performance comparison on robotic grasping dataset across different point cloud resolutions. OOM = Out of Memory. All models trained with rotation, permutation, and scaling augmentation.

| Model | Year | Rotation Error | | | | Distance Error | | | | Depth Error | | | | Normal Error | | | |
|---|---|---|---|---|---|---|---|---|---|---|---|---|---|---|---|---|---|
| | | 512 | 1024 | 2048 | 4096 | 512 | 1024 | 2048 | 4096 | 512 | 1024 | 2048 | 4096 | 512 | 1024 | 2048 | 4096 |
| DGCNN | 2018 | 0.015 | 0.019 | 0.031 | 0.120 | 3.01 | 5.34 | 8.34 | 10.30 | 0.08 | 0.08 | 0.09 | 0.08 | 0.013 | 0.017 | 0.023 | 0.051 |
| PointNet++ | 2017 | 0.015 | 0.021 | 0.028 | 0.080 | 2.51 | 4.88 | 5.57 | 9.54 | 0.08 | 0.08 | 0.07 | 0.09 | 0.013 | 0.018 | 0.021 | 0.048 |
| SEGNN | 2021 | 0.018 | 0.024 | 0.031 | 0.100 | 4.02 | 5.26 | 8.37 | 10.54 | 0.08 | 0.09 | 0.08 | 0.09 | 0.015 | 0.023 | 0.025 | 0.052 |
| SE(3)-Trans. | 2020 | 0.025 | 0.028 | OOM | OOM | 5.03 | 7.91 | OOM | OOM | 0.08 | 0.09 | OOM | OOM | 0.025 | 0.035 | OOM | OOM |
| Ours w/o local | 2025 | 0.019 | 0.025 | 0.030 | 0.090 | 4.02 | 5.10 | 8.30 | 9.85 | 0.08 | 0.08 | 0.08 | 0.08 | 0.013 | 0.017 | 0.020 | 0.041 |
| **Ours** | 2025 | **0.013** | **0.017** | 0.025 | **0.080** | **2.44** | **3.51** | **5.31** | **9.39** | 0.08 | 0.08 | **0.07** | 0.08 | **0.011** | **0.015** | **0.019** | **0.039** |

We benchmarked each of the baselines methods using the same model parameters as model net classification. Harmonics and nearest neighbors were chosen to be the max amount that fit on memory. Each training run was done on 8 NVIDIA v100 GPUs for 500 epochs.

## 6 Conclusion

**Conclusions.** This work tackles two key challenges in equivariant transformers: achieving scalability to global geometric context and efficiently computing high-order irreducible representations for greater expressiveness. By extending vector long convolution to Clebsch–Gordon convolution, we propose the first architecture that achieves global token attention in $\mathcal{O}(N \log N)$ time and supports arbitrary orders of irreducible representations. In addition, our method allows for permutation equivariance, which is essential for point cloud and atomic systems. We also provide comprehensive theoretical analyses to prove both equivariance and time complexity. Finally, we benchmark our method on various dataset, demonstrating clear gains in memory, speed, and accuracy across robotics, physics, and chemistry, outperforming existing state-of-the-art equivariant transformers.

**Limitations.** The $\mathcal{O}(L^3)$ complexity of our method is still suboptimal and may be prohibitive for cases requiring very high angular resolution.

**Future work.** Equivariant transformers can draw significant inspiration from computational astrophysics methods designed to efficiently handle $N$-body interactions.

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
