# OpenReview forum: "Fast and Global Equivariant Attention"
_ICLR.cc/2026/Conference — Submitted to ICLR 2026_

### Official Review · Reviewer_YhRc · 2025-10-29

**Soundness:** 2
**Presentation:** 3
**Contribution:** 2
**Rating:** 2
**Confidence:** 4

**Summary:**

This paper proposes the Clebsch-Gordan Transformer on top of the original SE(3)-Hyena, which fixes the missing permutation equivariance and enables its application to higher-order irreducible representations.

**Strengths:**

I have read the theoretical derivation in this article and believe there are no problems with it; I generally believe it is correct.

**Weaknesses:**

The shortcomings of this article are numerous and can be summarized as follows.

First, the paper claims to provide open-source code at https://anonymous.4open.science/r/CG-Transformer-0D1C/. However, no implementation of CG-Transformer can be found there. The repository appears to contain code related to SE(3)-HyperHyena, but even that lacks the core model components. Such a practice—appearing to open-source code while effectively withholding it—raises serious concerns about the reproducibility of the results and undermines the credibility of the work.

Second, the motivation of the model is not entirely convincing. The authors claim two primary contributions: (1) reducing the complexity of attention, and (2) reducing the complexity of tensor product operations. Both claims are problematic. The E2Former model has already demonstrated that tensor product computations can be localized to nodes, and with global attention, the overall complexity remains linear. Moreover, it remains unclear whether tensor product operations genuinely enhance performance. Recent spherical-scalarization models (e.g., ViSNet, HEGNN, GotenNet) have shown that relying solely on inner products—whose complexity is only $\mathcal{O}(L^2)$—is sufficient to achieve strong results. The tensor product design here therefore feels more like “innovation for its own sake” rather than a justified modeling necessity.

Third, the paper fails to engage adequately with relevant prior work. Key related models such as VN-EGNN, FastEGNN (which consider global contributions), as well as recent equivariant Transformers like GotenNet and eSEN, are not discussed. This omission weakens the contextual grounding of the paper’s contributions.

Finally, the experimental evaluation is insufficient. The paper omits important baselines, such as GotenNet and EquiformerV2 on QM9, and CGENN and its variants on N-body systems. For a paper emphasizing time complexity, the experiments are limited to small-scale datasets. A convincing demonstration would require evaluations on large-scale systems such as OC20 or MPtrj, where the claimed efficiency gains could be properly validated.

**Questions:**

See Weakness.

---

> ### Author Response · Authors · 2025-12-03
> **Response to Reviewer YhRc**
>
> We thank the reviewer for their comments. We respond point by point below:
>
> 1. "First, the paper claims to provide open-source code at https://anonymous.4open.science/r/CG-Transformer-0D1C/. However, no implementation of CG-Transformer can be found there. The repository appears to contain code related to SE(3)-HyperHyena, but even that lacks the core model components."
>
> Please check the repo again. The code is available. There was an issue (honest mistake) with the initial push, but the code is visible and to the best of our knowledge was available at the time of reviewing.
>
> 2. "Second, the motivation of the model is not entirely convincing. The authors claim two primary contributions: (1) reducing the complexity of attention, and (2) reducing the complexity of tensor product operations. Both claims are problematic. The E2Former model has already demonstrated that tensor product computations can be localized to nodes, and with global attention, the overall complexity remains linear."
>
> The E2Former method is again a local method. The goal of this paper was to develop a global method of equivariant attention (i.e. able to capture long rang correlations, which is, for example, critical for understanding critical phenomena in physics).
>
> 3. "Moreover, it remains unclear whether tensor product operations genuinely enhance performance. Recent spherical-scalarization models (e.g., ViSNet, HEGNN, GotenNet) have shown that relying solely on inner products—whose complexity is only $\mathcal{O}(L^2)$—is sufficient to achieve strong results. The tensor product design here therefore feels more like “innovation for its own sake” rather than a justified modeling necessity."
>
> We note that this in in disagreement with the first two reviewers, who state that:
> Reviewer TDu2: "The sparsity-centric implementation of CG-product is highly advantageous for the large-scale applications where higher-order features are desirable."
> Reviewer EptU: "The proposed approach is essentially a natural generalization of SE(3)-Hyena to higher-order irreducible representations,"
> However, we will benchmark against the recent inner product based methods, as it is not obvious to us which class of methods will be better.
>
>
> 3. "Third, the paper fails to engage adequately with relevant prior work. Key related models such as VN-EGNN, FastEGNN (which consider global contributions), as well as recent equivariant Transformers like GotenNet and eSEN… A convincing demonstration would require evaluations on large-scale systems such as OC20 or MPtrj, where the claimed efficiency gains could be properly validated."
>
> We actually agree with the reviewer on this point. We will benchmark on OC20 and MPtrj, and we will run more ablations comparing our method with VN-EGNN, FastEGNN, GotenNet and eSEN.

---

### Official Review · Reviewer_GpCg · 2025-10-31

**Soundness:** 2
**Presentation:** 2
**Contribution:** 2
**Rating:** 2
**Confidence:** 4

**Summary:**

This paper introduces the Clebsch–Gordan Transformer, a method for implementing fast, global SO(3)-equivariant attention.
The key idea is to extend SE(3)-Hyena  to arbitrary irreducible representations by replacing the vector cross product with a CG tensor product. The resulting CG convolution allows the model to process higher-order features beyond scalars and type-1 tensors.
The authors also exploit the sparsity structure of CG coefficients to reduce computational complexity to $O(L^3)$, and employ FFT/iFFT along the token dimension to achieve $O(N log N)$ scaling.

**Strengths:**

1. The paper addresses a relevant and challenging question: achieving global and high-order equivariant attention efficiently.
2. The exploration to the sparsity of CG coefficient is interesting.

**Weaknesses:**

1. The main contribution—replacing SE(3)-Hyena’s vector cross product with general Clebsch–Gordan tensor products—is conceptually straightforward and incremental. The proposed approach is essentially a natural generalization of SE(3)-Hyena to higher-order irreducible representations, rather than a fundamentally new mechanism for equivariant attention.
2. Although the authors claim that the CG Transformer is fast and efficient, there are no experiments demonstrating performance on long sequences or tasks with high-order input/output features. Such experiments would be necessary to substantiate the paper’s claim of scalability and efficiency.
3. The reported results on QM9, N-Body, and ModelNet40 are not particularly strong compared to established baselines such as EGNN or SEGNN. Many more recent equivariant architectures achieve higher accuracy or lower energy prediction error, which weakens the empirical significance of this work. I would suggest authors to explore other benchmarks as these ones are more or less saturated.

**Questions:**

Since the proposed method is more or less built upon SE(3)-Hyena, it would be important to provide direct comparisons with SE(3)-Hyena across all reported benchmarks.  Specifically, it would be nice if the paper include detailed results comparing (a) model accuracy, (b) computational efficiency (runtime), and (c) memory consumption. This would clearly demonstrate whether the proposed Clebsch–Gordan extension offers practical advantages by supporting higher-order representations.

**Details Of Ethics Concerns:**

None.

---

> ### Author Response · Authors · 2025-12-03
> **Response to reviewer GpCg**
>
> We thank the reviewer for their comments. We address reviewer concerns point by point:
>
> Weaknesses:
> 1. "The main contribution—replacing SE(3)-Hyena’s vector cross product with general Clebsch–Gordan tensor products—is conceptually straightforward and incremental. The proposed approach is essentially a natural generalization of SE(3)-Hyena to higher-order irreducible representations, rather than a fundamentally new mechanism for equivariant attention."
>
> The assumption that the generalization of the cross product is "conceptually straightforward" is a subjective statement. However, we believe that the use of the Fourier space to achieve permutation equivariance is a non-trivial, and novel, contribution.
>
>
> 2. "Although the authors claim that the CG Transformer is fast and efficient, there are no experiments demonstrating performance on long sequences or tasks with high-order input/output features. Such experiments would be necessary to substantiate the paper’s claim of scalability and efficiency.The reported results on QM9, N-Body, and ModelNet40 are not particularly strong compared to established baselines such as EGNN or SEGNN."
>
> We recommend the reviewer look at the "Object Grasping" test, where other methods OOM as the number of points increase up to N=4096. Our method does not OOM on this large point cloud, and has essentially constant error as the number of points is further increased. Furthermore, for the "Nbody" experiments we compare the SE(3)-Hyena directly with our method.
>
>
> Questions:
> "Since the proposed method is more or less built upon SE(3)-Hyena, it would be important to provide direct comparisons with SE(3)-Hyena across all reported benchmarks. Specifically, it would be nice if the paper include detailed results comparing (a) model accuracy, (b) computational efficiency (runtime), and (c) memory consumption. This would clearly demonstrate whether the proposed Clebsch–Gordan extension offers practical advantages by supporting higher-order representations."
>
> We agree with the reviewer on this point. However, we would refer the reader to the appendix, specifically the "Object Grasping" experiments where we conduct numerous experiments comparing our method with SE(3)-Hyena in terms of time, memory and speed.

---

### Official Review · Reviewer_EptU · 2025-10-31

**Soundness:** 2
**Presentation:** 2
**Contribution:** 2
**Rating:** 2
**Confidence:** 5

**Summary:**

This paper generalizes the SE(3)-Hyena method from Moskalev et al. 2024 to support higher order spherical harmonics and efficent Clebsh-Gordan transforms to combine them.

**Strengths:**

The problem considered is relevant and the proposed solution is potentially impactful.
In this sense, the paper is well motivated and the pursued direction very interesting.
Unfortunately, I see many major flaws in the current manuscript.

**Weaknesses:**

Unfortunately, I think the current manuscript has a few major issues.

First of all, it is claimed throughout the paper that this work improves SE(3)-Hyena by maintaining its O(N log N) complexity while also preserving permutation equivariance (a key property lost in SE(3)-Hyena). I think this could be the most impactful contribution of the paper. Unfortunately, the main paper never explains how this is achieved.
Only in Appendix C (never referred to from the main paper), it is explained this is done by either:

  1) using the Graph Fourier Transform (instead of Hyena's FFT), which preserves the permutation equivariance. However, as far as I know, this also comes with a O(N^3) scaling to compute the eigendecomposition of the graph Laplacian, making this solution much less scalable than the attention baseline. Unfortunately, this is never discussed.

  2) simply adopting SE(3)-Hyena's FFT-based solution together with data augmentation, which only provides approximate equivariance.

Hence, I feel like the claims in the main paper are quite misleading.

Moreover, Sec 5.2 and 5.3 test the scalability of the proposed model on relative small graphs (N<=40 in 5.2 and N<=9 in 5.3). While the results are promising, this doesn't seem like a suitable benchmark to demonstrate the high scalability of the proposed method to large N. As a side note, if I remember correctly, model performance on QM9 benchmark has essentially saturated, so I am not sure it is a great benchmark anymore. Results on larger graphs like ModelNet in Sec 5.4 are instead less promising.

This makes me wonder if replacing a standard transformer in favor of the proposed method and, therefore, trading off the permutation equivariance of standard attention and enforcing SO(3) equivariance, is really a good design strategy. Indeed, the permutation group can be extremely large (N!), while the rotation group SO(3), despite being theoretically infinite, acts very smoothly on the features and is only a 3-dimenisonal Lie group; somehow, the N!-size permutation group seems to be much more relevant here.

**Questions:**

Line 177: Is $W_K^l$ supposed to be $W_K^{l, l'}$ ?

Sec 4.3: the bra-kets notation is introduced with no proper definiton. What does it mean $|lm\rangle$ is the basis for the $l$ representation"?
Sec 4.3: Soon later, what is $J^2$? Is that the tensor product of the $J$-th irrep with itself? What is $J_z$? It was never defined earlier
Sec 4.3: $C^{JM}_{lm, l'm'}$ never defined properly (only $Jll'$ indeces used before).

Fig 3: can you fix the image size and the font? It is very hard to read it currently

Line 402: "We 3 perform..." is there a typo maybe? The link seems also broken

Sec 5.4: results on ModelNet (which has many more nodes per graph) don't seem very promising, especially in ModelNet40, even when comparing with other equivairant baselines. Could the authors comment on that?

It is claimed in the main paper the somehow the permutation equviariance is preserved by the FFT-based attention but never really explained how. This seesm to be disucssed in Appendix C which is however never referred to from the main paper. Moreover, appendix C refers to appendix E for comparisons between numerous methods to enforce permutation equivariance but I couldn't find them.

---

> ### Author Response · Authors · 2025-12-03
> **Response to reviewer EptU**
>
> We thank the reviewer for their comments. We address reviewer concerns point by point:
>
> 1. " using the Graph Fourier Transform (instead of Hyena's FFT), which preserves the permutation equivariance. However, as far as I know, this also comes with a O(N^3) scaling to compute the eigendecomposition of the graph Laplacian, making this solution much less scalable than the attention baseline. Unfortunately, this is never discussed. simply adopting SE(3)-Hyena's FFT-based solution together with data augmentation, which only provides approximate equivariance."
>
> We used a Lanczos method to compute the lowest eigenvectors of the graph Laplacian. Because the graph Laplacian is positive definite and symmetric, the complexity scales like O(MN) where M=logN is the number of eigenvectors used. This is much lower than the naive O(N^3) diagonalization.
>
> 2. "Moreover, Sec 5.2 and 5.3 test the scalability of the proposed model on relative small graphs (N<=40 in 5.2 and N<=9 in 5.3). While the results are promising, this doesn't seem like a suitable benchmark to demonstrate the high scalability of the proposed method to large N. As a side note, if I remember correctly, model performance on QM9 benchmark has essentially saturated, so I am not sure it is a great benchmark anymore. Results on larger graphs like ModelNet in Sec 5.4 are instead less promising."
>
> We agree that QM9 is mostly saturated. We recommend the reviewer look at the "Object Grasping" test, where other methods OOM as the number of points increase up to N=4096. Our method does not OOM on this large point cloud, and has essentially constant error as the number of points is further increased.
>
> 3. "This makes me wonder if replacing a standard transformer in favor of the proposed method and, therefore, trading off the permutation equivariance of standard attention and enforcing SO(3) equivariance, is really a good design strategy. Indeed, the permutation group can be extremely large (N!), while the rotation group SO(3), despite being theoretically infinite, acts very smoothly on the features and is only a 3-dimenisonal Lie group; somehow, the N!-size permutation group seems to be much more relevant here."
>
> The whole point of this paper is to try and achieve a both SE(3) and permutation group equivariance, as this is the natural symmetry group of point clouds. We believe that our work archives both approximate SE(3) and permutation group equivariance, with "approximate" being due to truncation and numerical discretization.

---

### Official Review · Reviewer_TDu2 · 2025-10-31

**Soundness:** 3
**Presentation:** 2
**Contribution:** 3
**Rating:** 4
**Confidence:** 4

**Summary:**

The paper introduces the Clebsch-Gordan Transformer, a novel method for efficient global equivariant attention for geometric systems. It achieves this through Clebsch-Gordan long convolution, enabling log-linear scaling with context length and efficient tensor products through utilising the sparsity of the CG-product. The method also incorporates optional permutation equivariance via weight sharing or data augmentation. Experiments on tasks like n-body, QM9, ModelNet classification, and robotic grasping show improved performance and memory efficiency over some of the existing equivariant models.

**Strengths:**

1.	The context and computational efficiency of equivariant transformers severely limit their practical applicability and diminish the added value of equivariance by significantly increasing training and deployment costs. The paper thus tackles a very important problem and addresses a key limitation of modern equivariant self-attention architectures.
2.	The sparsity-centric implementation of CG-product is highly advantageous for the large-scale applications where higher-order features are desirable. Especially, given that the original work of (Moskalev et al. 2025) Geometric Hyena Networks, does not focus on sparse implementation of steerable long convolutions.
3.	The method potentially allows incorporating limited permutation equivariance to the convolutional mechanism, which makes it applicable to a broader class of geometric systems.
4.	The proposed method is tested in a diverse array of tasks, ranging from molecules to robotics, and experimental results support the advantages of the proposed method. In addition, the paper provides extensive ablations on most of the design choices.

**Weaknesses:**

1.	The first contribution of the work is “generalise SE(3)-Hyena method of (Moskalev et al., 2024) to include equivariant features of all types”. This is already outlined in the original follow-up work, see Appendix A5 in Geometric Hyena Networks (Moskalev et al., 2025). With this, the generalisation of SE(3)-Hyena to equivariant features of all types can not be claimed as a novel and original contribution of this work. This should be made clear throughout the paper.
2.	The paper never refers Geometric Hyena Networks, which outlines steerable long-convolutional mechanism that enables all equivariant types.
3.	Although the method is designed to tackle large-scale equivariant learning, the presented experiments are rather small-scale. N-body: 40 particles, QM9: up to 9 heavy atoms, ModelNet: up to 1024 points. Only a moderate-scale experiment in the paper is Object Grasping with up to 4096 points, which is still far away from the scale where log-linear significantly outpaces quadratic complexity in terms of runtime. This creates a mismatch where paper positions itself as the method for equivariant-learning tasks that require context and compute efficiency, but only provides experiments on relatively small-scale datasets.
4.	The experimental comparison against existing equivariant methods is very limited. The paper only test the proposed approach against SE(3)-Transformer, SEGNN, EGNN, and EGNN among equivariant methods.
5.	The permutation equivariance section should be in the main text, as it is a significant contribution on top of long convolutions for point clouds.
6.	The paper will significantly benefit from wall clock runtime and memory usage analysis (over the context length) of the proposed model over the existing methods.

**Questions:**

1. The introduction states the computational complexity of NlogN, but related work (096-097) says that the proposed method is linear time. Is it a typo? Should it be log-linear?
2. The appendix mentions data augmentation with regularisation as one of the optimal ways to achieve permutation equivariance, but does not provide any further details. Can authors elaborate on this?
3. The proof of permutation equivariance for Fourier space attention assumes that the Laplacian eigenvectors transform canonically under permutations. Does this hold as well in cases of eigenvalue degeneracy?

---

> ### Author Response · Authors · 2025-12-03
> **Response to reviewer TDu2**
>
> We thank the reviewer for their thorough review. We respond to the reviewer criticisms below:
>
> 1. "The first contribution of the work is “generalise SE(3)-Hyena method of (Moskalev et al., 2024) to include equivariant features of all types”. This is already outlined in the original follow-up work, see Appendix A5 in Geometric Hyena Networks (Moskalev et al., 2025). "
>
> There are two versions of the SE(3)-Hyena paper. The original paper contained only invariant and vector features. To the best of our knowledge at the time of writing this paper the second version of the SE(3)-Hyena was not public.
>
> 2. "The paper never refers Geometric Hyena Networks, which outlines steerable long-convolutional mechanism that enables all equivariant types."
>
> We were unaware of this work. We thank you for bringing this paper to our attention and we will benchmark and cite this work in future versions of this paper.
>
> 3. "Although the method is designed to tackle large-scale equivariant learning, the presented experiments are rather small-scale. N-body: 40 particles, QM9: up to 9 heavy atoms, ModelNet: up to 1024 points. Only a moderate-scale experiment in the paper is Object Grasping with up to 4096 points, which is still far away from the scale where log-linear significantly outpaces quadratic complexity in terms of runtime."
>
> We are a bit confused about this criticism. For the Object Grasping test, we increased the number of point until the point where all baseline method OOM'd. We will run larger tests in the future, but the tests performed in the paper are at least to the memory usage of most current baseline methods.
>
>
> 4. "The permutation equivariance section should be in the main text, as it is a significant contribution on top of long convolutions for point clouds."
>
> We appreciate the recognization of the importance of permutation invariance as a strength. We will move this result to the main text for the next paper.
>
> 5. "The paper will significantly benefit from wall clock runtime and memory usage analysis (over the context length) of the proposed model over the existing methods."
> Understood. We will add existing wall clock and memory usage analysis tests in the future.
>
> Questions:
> 1. "The introduction states the computational complexity of NlogN, but related work (096-097) says that the proposed method is linear time. Is it a typo? Should it be log-linear?"
>
> Yes this should say "Log-Linear Attention". Thank you for the correction.
>
> 2. "The appendix mentions data augmentation with regularisation as one of the optimal ways to achieve permutation equivariance, but does not provide any further details. Can authors elaborate on this?"
>
> Yes. When training, we simply randomly shuffled the order of points. We found that this was an effective way to reduce generalization error.
>
> 3. "The proof of permutation equivariance for Fourier space attention assumes that the Laplacian eigenvectors transform canonically under permutations. Does this hold as well in cases of eigenvalue degeneracy?"
>
> This is an excellent question. The proof still holds, with the caveat that instead of the vector being permutation equivariant, the projector into the linear subspace spanned by the degenerate eigenvalues is now permutation equivariant.

---

### Meta-Review · Area_Chair_i5xH · 2025-12-25

**Summary:**

The paper introduces Clebsch-Gordon Convolution to provide a more efficient equivariant attention operation for the SE(3) groups.
Several reviewers felt that the experiments provided in the current manuscript were not convincing enough to demonstrate the efficacy of the proposed method at more complex and challenging scales. The authors, in their rebuttal, point to the object grasping experiment as one that demonstrates the efficiency of their approach over baselines. However, the baselines do not encompass many more recent attempts at equivariant architectures, including SE(3)-Hyena and its follow-ups. This paper could demonstrate the promise by using equivariant convolution in areas that are on a much larger scale, such as protein backbones or materials design, where SE(3) equivariance is currently being abandoned.

With regards to the presentation, many reviewers also felt that a large portion of the appendix content was deserving of the main paper. Key details, such as using the Lanczos method to compute the eigenvalues of the graph laplacian and other important details, are delegated to the appendix. I recommend a more comprehensive rewrite of the paper following the reviewers' suggestions. Furthermore, reviewer TDu2 raised valid points regarding the novelty of this work in comparison to Geometric Hyena networks. This should be updated in the paper as concurrent or related work.

Given these points, the recommendation for this paper is a rejection.

**Reviewer Concerns:**

## Points addressed

- Points regarding computing cost of the graph Laplacian
- Clarification on equivariance being approximate

## Points outstanding
- Missing baselines that the authors themselves acknowledged as missing
- Missing larger-scale experiments beyond object grasping
- Polishing of the main text in relation to appendix content.

**Reviewer Scores:**

None of the reviewers would have likely increased their scores.

---

### Decision · Program_Chairs · 2026-01-26

Reject